# Unlocking Decoder-LLMs for Text Embedding with Instructions, Soft Supervision and Curriculum Learning

## Abstract

Large language models (LLMs) are increasingly used for text embedding, yet most decoder-only architectures remain underexplored for this purpose. We present a unified instruction-based framework that adapts decoder-only LLMs into general-purpose text encoders without architectural modifications. Our approach integrates four complementary techniques: (i) in-context learning with structured instructions to generate context-aware embeddings without costly fine-tuning, (ii) soft supervision via knowledge distillation from a high-performance teacher retrieval pipeline, (iii) adaptive margin-based hard-negative mining to stabilize contrastive learning, and (iv) a principled two-stage curriculum learning strategy that first builds a semantic foundation on Semantic Textual Similarity (STS) before specializing on retrieval tasks.

Our analysis shows that this sequential curriculum is critical for robust performance, substantially outperforming simultaneous multi-task training. Evaluated on the 41 diverse tasks of the MTEB (English, v2) benchmark, our model achieved the state-of-the-art results, and consistently ranks among the very top models demonstrating both strong overall performance and robustness compared to larger or fully fine-tuned models. Notably, it excels in semantically demanding categories such as Retrieval, Semantic Textual Similarity, and Summarization. These results highlight the effectiveness of strategically combining instruction-based prompting, soft-label distillation, adaptive sampling, and curriculum learning to unlock the potential of decoder-only LLMs as powerful and flexible text embedding models.

## 1 Introduction

Text embeddings are a cornerstone of modern NLP, and the advent of large language models (LLMs) has catalyzed a shift toward using them as powerful, general-purpose text encoders. Models like Llama2Vec (BehnamGhader et al., 2024), E5-Mistral (Wang et al., 2023) and NV-Embed (Lee et al., 2024a) have demonstrated remarkable performance by fine-tuning decoder-only LLMs. However, these successes often hinge on full-parameter fine-tuning or significant architectural modifications, which can be computationally expensive and may constrain the models' intrinsic generalization abilities. Furthermore, the efficacy of contrastive learning, a standard training paradigm, is often hampered by supervision quality, including noisy labels and the presence of false negatives.

To address these limitations, we propose a unified framework designed to generate high-quality embeddings by harnessing the latent strengths of LLMs with minimal intervention. Our approach is built on several synergistic innovations. First, we leverage in-context learning (ICL), guiding the model with task-specific instructions and few-shot examples to produce specialized embeddings without updating its weights, thus maximizing flexibility and efficiency. Second, we tackle the supervision quality problem by incorporating soft labeling, distilling continuous-valued relevance scores from a state-of-the-art teacher retrieval pipeline. These nuanced signals provide richer semantic guidance than traditional binary labels. Third, we introduce an adaptive margin-based strategy for hard-negative mining, which dynamically filters out ambiguous examples to stabilize training and sharpen the model's discriminative power.

Beyond what the model learns from, we identify that how the model learns is equally critical. Our investigation into curriculum learning reveals that a structured, sequential training strategy is a key driver of performance. We demonstrate that a two-stage curriculum, which first establishes a strong semantic foundation with a Semantic Textual Similarity (STS) task before specializing on a discriminative retrieval task, is significantly more effective than conventional multi-task learning.

This paper makes the following key contributions:

- We present a unified, instruction-tuned framework that synergistically combines in-context learning, soft-label distillation, and adaptive negative sampling to train a powerful text embedder from a frozen decoder-only LLM.

- We demonstrate that knowledge distillation from a hybrid teacher pipeline, which provides continuous relevance scores, significantly improves the semantic quality of embeddings over training with hard labels alone.

- Through extensive ablation studies, we provide a systematic analysis of curriculum learning, revealing that a two-stage curriculum—first training on semantic similarity (STS) and then on retrieval tasks—is critical for achieving balanced, state-of-the-art performance and substantially outperforms simultaneous multi-task learning.

- Our proposed model achieves a top-tier Borda rank on the comprehensive MTEB benchmark, validating its robust generalization across 41 tasks and outperforming strong baselines in key areas like retrieval and summarization.

## 2  RELATED WORKS

**LLM-based Text Embedding Models**   Recent research has increasingly explored the use of large language models (LLMs) as backbone encoders for text embedding tasks. This shift is evident in models such as Llama2Vec(BehnamGhader et al., 2024), which introduced two pretraining objectives to better align LLMs with embedding tasks, yielding substantial performance improvements on retrieval benchmarks like BEIR. However, its performance on the MTEB leaderboard remains relatively modest. Other models such as E5-Mistral(Wang et al., 2023), Linq(Kim et al., 2024), and Gecko(Lee et al., 2024b) have leveraged large-scale synthetic data to effectively fine-tune LLMs, achieving strong results across both retrieval and non-retrieval tasks. NV-Embed(Lee et al., 2024a) further advances this line of work by incorporating a latent attention pooling mechanism and a two-stage training strategy to mitigate false negatives, leading to significant improvements in retrieval robustness. In contrast, our framework achieves stronger performance without architectural changes or full fine-tuning. By preserving the frozen backbone and relying on instruction-driven embedding generation, we maximize efficiency and retain the LLM's inherent generalization capacity.

**Shift Toward In-Context Learning**   Despite the strong performance of LLM-based embedding models, prior approaches have often relied heavily on architectural modifications—such as replacing unidirectional attention with bidirectional mechanisms—or full model fine-tuning. These strategies, while effective, tend to overlook the inherent generalization capabilities of LLMs and require substantial computational resources. Recently, however, there has been a shift toward leveraging in-context learning (ICL) as a more efficient alternative. Models such as BGE-en-icl (Li et al., 2024) demonstrate that task-specific prompts and demonstrations can be used to condition LLMs for embedding generation without modifying model weights. This emerging paradigm highlights the potential of ICL for building flexible and adaptive embedding systems that generalize well across tasks while minimizing the cost of training and deployment.

**High-Quality Supervision via Soft Labeling and Hard-Negative Mining.**   Recent advances in embedding model training highlight the importance of soft labeling and hard-negative mining. In particular, soft supervision signals derived from high-capacity reranker models—often implemented via teacher-student frameworks—have proven effective in guiding embedding models toward better alignment with semantic similarity objectives (Mandal et al., 2024). Meanwhile, hard-negative mining strategies play a critical role in closing the semantic gap between positive and negative pairs and in mitigating the risk of false negatives during contrastive learning. Building on these insights, our goal is to develop a generalized embedding model that preserves the inherent strengths of LLMs

by leveraging in-context learning. Without requiring architectural modifications or full-scale fine-tuning, our approach aims to achieve strong adaptability across both retrieval and non-retrieval tasks, while maintaining efficiency in training and deployment.

## 3 TRAINING DATASET

**Retrieval Datasets**   Following the common approaches taken by top-performing models on the English MTEB leaderboard (Enevoldsen et al., 2025), we employ a set of publicly available retrieval datasets, including MSMARCO (Nguyen et al., 2016), HotpotQA (Yang et al., 2018), Natural Questions (Kwiatkowski et al., 2019), SQuAD (Rajpurkar et al., 2016), ELI5 (Fan et al., 2019), ArguAna (Wachsmuth et al., 2018), FiQA (Maia et al., 2018), FEVER (Thorne et al., 2018), Quora Duplicate Questions (Sharma et al., 2019). These retrieval datasets are annotated not only with hard labels, such as positive and negative pairs, but are also enhanced with soft labels, which are described in detail in the following section 4.1. Furthermore, they are processed through a sophisticated hard-negative mining strategy, which will also be detailed in the subsequent section 4.3.

**Non-Retrieval Datasets**   Similar to other models (Lee et al., 2024a; Li et al., 2024), we also incorporate publicly available datasets from non-retrieval benchmarks, particularly those associated with classification, clustering, reranking, and semantic textual similarity (STS) within the MTEB tasks. Importantly, we use only the training data and omit test splits to maintain evaluation integrity.

For classification tasks, we utilize a range of benchmark datasets, including AmazonCounterfactual-Classification (O'Neill et al., 2021), AmazonReviewsClassification (McAuley & Leskovec, 2013), Banking77Classification (Casanueva et al., 2020), EmotionClassification (Saravia et al., 2018), ImdbClassification (Maas et al., 2011), MTOPIntentClassification (Li et al., 2020), ToxicConversationsClassification (cjadams et al., 2019), and TweetSentimentExtractionClassification (Maggie et al., 2020).

We utilize a range of clustering datasets such as ArxivClustering[1], BiorxivClustering [2], MedrxivClustering[3], TwentyNewsgroupsClustering (Lang, 1995), RedditClustering (Geigle et al., 2021), StackExchangeClustering (Geigle et al., 2021).

For reranking tasks, we incorporate SciDocsRR (Cohan et al., 2020) and StackOverflowDupQuestions (Liu et al., 2018) into our training corpus. Additionally, we include the training splits of three widely used semantic textual similarity datasets—STS12 (Agirre et al., 2012), STS22 (Chen et al., 2022), and STS-Benchmark (Chen et al., 2022)—to enhance the model's ability to capture fine-grained semantic relations.

Overall, our training corpus covers retrieval, classification, clustering, reranking, and STS tasks, providing diverse supervision to improve generalization. A full list of task-specific instruction templates is provided in Appendix B.2.

**Data Conversion**   To unify heterogeneous supervision across tasks, we convert certain datasets into a common instruction-response format. In particular, natural language inference (NLI) data are reformulated into semantic textual similarity (STS) style sentence pairs, allowing the model to learn fine-grained semantic relationships in a consistent way across tasks. Pairs labeled as *entailment* are converted to positive pairs with high similarity scores, and *contradiction* pairs are converted to negative pairs, while pairs labeled as *neutral* are excluded from training due to their ambiguous semantic alignment. Further details of this conversion process, including the mapping strategy and examples, are provided in Appendix B.3.

## 4 PROPOSED METHODS

Our framework integrates multiple components including a hybrid retrieval pipeline, soft-label distillation, instruction-based embedding generation, and curriculum learning. To provide an overview of how these components interact, we illustrate the overall architecture in Figure 1.

---

[1]https://www.kaggle.com/datasets/Cornell-University/arxiv

[2]https://api.biorxiv.org/

[3]https://api.medrxiv.org/

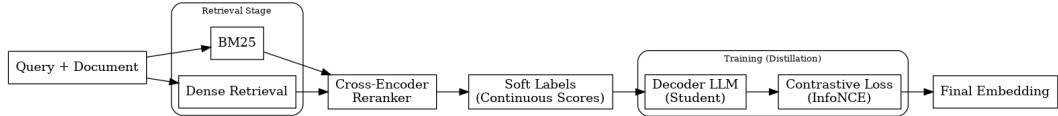

Figure 1: Overall architecture of our framework. A hybrid retrieval and reranking pipeline produces soft labels, which supervise the training of a decoder-only LLM via contrastive learning, yielding the final embedding model.

## 4.1 KNOWLEDGE DISTILLATION VIA SOFT LABELING

To enhance the semantic discrimination capability of our embedding model, we employ soft-label distillation using similarity scores generated by our retrieval pipeline system. These continuous-valued relevance signals serve as soft targets, guiding the embedding model to learn nuanced semantic relationships through contrastive learning. We opted for this pipeline-based approach as it provides a strong balance of high performance and computational efficiency, especially when compared to more resource-intensive alternatives like large language models (LLMs). A more detailed discussion on this design choice and future directions is presented in Section 6.

Unlike hard labels, which provide binary relevance judgments, soft labels offer richer information by capturing the teacher model's confidence across different candidates. This approach aligns with recent theoretical insights: Mandal et al. (Mandal et al., 2024) demonstrate that soft label supervision allows student models to generalize more effectively and requires fewer neurons to approximate the teacher's decision boundaries compared to hard-labeled training targets.

For soft-label distillation in this study, we employ a two-stage retrieval pipeline, as illustrated in Figure 2. The pipeline integrates both lexical and semantic retrieval components, followed by a reranker to generate relevance scores. In the first stage, candidate passages are retrieved using both lexical search and semantic retrieval. In the second stage, the retrieved candidates are re-ranked using the reranker, which assigns fine-grained relevance scores based on contextual alignment with the query. The final ranked list is used for downstream tasks or soft-label distillation.

We employ three scoring functions to estimate the relevance between a query $q$ and a document $d$:

- **Lexical Search (BM25):**
  The BM25 score is computed based on exact token overlap, using the formula:

$$\text{Lexical score}(d, q) = \sum_{i=1}^{n} \text{IDF}(q_i) \cdot \frac{f(q_i, d) \cdot (k_1 + 1)}{f(q_i, d) + k_1 \cdot \left(1 - b + b \cdot \frac{\text{len}}{\text{avglen}}\right)}$$

- **Semantic Search (Dense Retrieval):**
  Semantic similarity is estimated via the dot product between dense query and document embeddings:

$$\text{Semantic score}(q, d) = q \cdot d = \sum_{i=1}^{n} q_i \cdot d_i$$

- **Reranker (Cross-Encoder):**
  Fine-grained semantic relevance is captured by a cross-encoder model $f_\theta$, which takes both query and document as input:

$$\text{Reranker score}(q, d) = f_\theta([\text{CLS}], q, d)$$

To integrate the individual rankings from these components—denoted as $L$ (lexical), $S$ (semantic), and $R$ (reranker)—we adopt the Reciprocal Rank Fusion (RRF) algorithm (Cormack et al., 2009). The combined score for a document $d$ is computed as:

$$\text{RRF}_{\text{score}_{\text{total}}} = \sum_{l \in L} \frac{1}{k + l(d)} + \sum_{s \in S} \frac{1}{k + s(d)} + \sum_{r \in R} \frac{1}{k + r(d)}$$

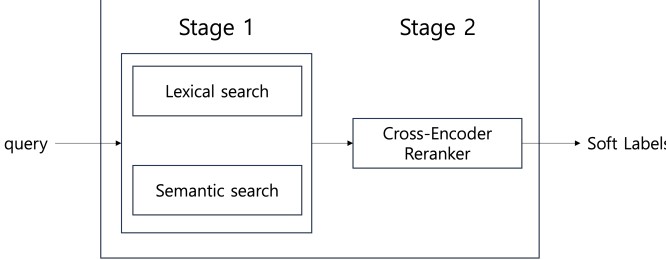

Figure 2: Overview of the two-stage retrieval pipeline for soft labeling. In the first stage, candidate passages are retrieved using both lexical and semantic search. The second stage employs a cross-encoder reranker to refine the results by scoring query-document pairs for final ranking, which are then used as soft labels.

where $k$ is a constant hyperparameter, typically set to 60 as recommended by the original paper.

In our implementation, we adopt the standard InfoNCE loss (Izacard et al., 2021) as the contrastive training objective. Given a batch of queries $q_i$, their corresponding positive passages $p_i$, and a group of hard negatives corresponding to each query $\{n_{i,j}\}$, the loss is calculated as:

$$\mathcal{L} = -\sum_{i=1}^{N} \log \frac{\exp(\text{sim}(q_i, p_i)/\tau)}{\sum_{j=1}^{N} \exp(\text{sim}(q_i, n_{i,j})/\tau) + \exp(\text{sim}(q_i, p_i)/\tau)}$$

where $\text{sim}(\cdot)$ denotes cosine similarity and $\tau$ is a temperature parameter.

Using soft targets from a strong teacher model, our embedding model is able to internalize fine-grained semantic cues encoded in teacher output, leading to improved retrieval performance and robustness across multiple tasks.

## 4.2 CONTEXT-AWARE EMBEDDING GENERATION VIA STRUCTURED INSTRUCTIONS

Traditional embedding models typically generate representations by directly encoding input text using an encoder or a language model. While this approach is simple and computationally efficient, it lacks the flexibility to capture diverse user intents and domain-specific nuances, thereby limiting generalization capabilities. In contrast, large language models (LLMs) with decoder only, such as Mistral-7B (Jiang et al., 2023), have shown strong abilities to utilize contextual information from structured inputs. In particular, decoder-only LLMs possess strong in-context learning (ICL) capabilities, and a recent study has explored fine-tuning methods that effectively exploit the in-context learning capabilities of decoder-only LLMs (Li et al., 2024).

In our approach, we enable context-aware embedding generation by incorporating structured instructions along with few-shot examples. Each query is augmented with a task-specific instruction and a set of demonstration pairs (query-passage), designed to simulate the semantics of the target task. The input sequence is terminated with an [EOS] token, and the final embedding is extracted from the representation of the last token. This design allows the model to implicitly learn task formats, intents, and output patterns without requiring any parameter updates.

Formally, given a task instruction $t$, support examples $\{(q_i, p_i)\}_{i=1}^{k}$, and a target query $q$, the input is formatted as follows:

$$\text{Examples}\{(q_i, p_i)\}_{i=1}^{k} + \text{Instruct: } \{\text{task\_definition}\} + \text{Query}$$

The resulting embedding, extracted from the [EOS] token, reflects both the semantic content of the query and its contextual relevance to the task specification.

Table 1: Comparison of top-ranked models on the MTEB (English v2) benchmark. Scores are averaged across 41 tasks spanning retrieval, reranking, clustering, pair classification, classification, STS, and summarization. Models are ordered by Borda rank, and the results demonstrate competitive performance, particularly in retrieval and summarization. The reported scores correspond to the snapshot of the official leaderboard as of June 2025.

| Model | Retrieval | Reranking | Clustering | PairClassification | Classification | STS | Summarization | Mean (Task) |
|---|---|---|---|---|---|---|---|---|
| *# of tasks* | 10 | 2 | 8 | 3 | 8 | 9 | 1 | 41 |
| Seed1.5-Embedding | 67.45 | 50.67 | 60.83 | 87.39 | 89.88 | 87.23 | 36.44 | 74.76 |
| gemini-embedding-001 | 64.35 | 48.59 | 59.39 | 87.70 | 90.05 | 85.29 | 38.28 | 73.30 |
| Linq-Embed-Mistral | 60.14 | 49.44 | 54.07 | 88.44 | 83.00 | 84.69 | 37.26 | 69.80 |
| jasper_en_vision_language_v1 | 56.05 | 50.00 | 60.52 | 88.14 | 90.27 | 84.37 | 37.19 | 71.41 |
| SFR-Embedding-Mistral | 59.33 | 50.15 | 54.93 | 88.59 | 80.47 | 84.77 | 36.32 | 69.31 |
| NV-Embed-v2 | 62.84 | 49.61 | 47.66 | 88.69 | 87.19 | 83.82 | 35.21 | 69.81 |
| text-embedding-005 | 58.77 | 48.84 | 51.91 | 87.62 | 86.03 | 85.18 | 35.05 | 69.60 |
| text-embedding-004 | 59.06 | 48.48 | 51.52 | 87.65 | 86.03 | 84.84 | 36.12 | 69.53 |
| gte-Qwen2-7B-instruct | 58.09 | 50.47 | 58.97 | 85.90 | 88.52 | 82.69 | 35.74 | 70.72 |
| e5-mistral-7b-instruct | 57.62 | 49.78 | 51.44 | 88.42 | 79.85 | 84.32 | 36.57 | 67.97 |
| **Ours** | 66.18 | 49.13 | 59.25 | 88.67 | 89.97 | 86.69 | **38.93** | 74.12 |

## 4.3 ADAPTIVE MARGIN-BASED MINING STRATEGIES

Inspired by existing methodologies, we improve the robustness of contrastive learning and mitigate the impact of false negatives by applying adaptive margin-based mining strategies that select the negative passages of the top K according to the relevance score of the corresponding positive. This approach dynamically adjusts the threshold for negative selection, ensuring that semantically similar but nonidentical passages, often mislabeled as negatives, are excluded from training. As a result, it reduces noise and preserves the semantic contrast signal essential for effective embedding learning.

Adopting the strategy proposed by Moreira et al. (Moreira et al., 2024), we take advantage of the tow-stage IR pipeline as a teacher retrieval model to identify high-quality hard negative passages for each query. We define the maximum allowable score for negative passages as a fixed proportion of the corresponding positive score, applying a 95% margin. This is expressed as:

$$\text{max\_negative\_score\_threshold} = \text{positive\_score} \times \text{percentage\_of\_margin}$$

Negative candidates with scores falling below this threshold are excluded during training. Subsequently, a random subset is sampled from the top-K ranked negatives to promote training diversity and mitigate overfitting to commonly occurring distractors. These strategies are simple yet effective and can be flexibly combined. They collectively ensure that training focuses on hard negatives that are informative but not semantically indistinguishable from the positives—thereby enhancing both retrieval accuracy and training stability.

## 5 EXPERIMENTS

### 5.1 EXPERIMENTAL SETUP

**Backbone Model**  We use Mistral-7B (Jiang et al., 2023) as our backbone model, initialized with the official pre-trained weights. This choice is consistent with recent state-of-the-art text embedding models such as E5-Mistral (Wang et al., 2023) and NV-Embed-v2 (Lee et al., 2024a).

**Fine-tuning Strategy**  We adopt parameter-efficient fine-tuning using Low-Rank Adaptation (LoRA) (Hu et al., 2022), combined with in-batch negative sampling that incorporates multiple hard negatives for retrieval-oriented tasks. Detailed hyperparameter settings, including LoRA configuration, learning rate schedule, and warm-up strategy, are provided in Appendix B.1.

**Evaluation Benchmark**  We evaluate the model on the MTEB (English v2) benchmark, which consists of 41 tasks across seven categories, including classification, clustering, retrieval, reranking, and STS. To enable fair comparison, we follow the official Borda count ranking protocol (Colombo et al., 2022; Enevoldsen et al., 2025).

## 5.2 MAIN RESULTS ON MTEB BENCHMARK

Table 1 presents the average performance across seven task categories—*classification*, *clustering*, *pair classification*, *re-ranking*, *retrieval*, *semantic textual similarity (STS)*, and *summarization*—comparing our model against state-of-the-art methods reported on the MTEB leaderboard snapshot as of mid-2025.

While the model marked as *Ours* in Table 1 achieves a mean score of 74.12, which is competitive with the top-ranked system (74.76), it obtains a higher overall Borda rank. This indicates that the model delivers more consistent and robust performance across a diverse spectrum of tasks, rather than being specialized for a narrow set of categories.

Notably, our model achieves top-tier results in categories that demand a deep understanding of semantic nuances, including Retrieval (66.18; 2nd place), STS (86.69; 2nd place), and Pair Classification (88.67; 2nd place). Furthermore, it secured the highest score (38.93) among all evaluated systems in the Summarization task, demonstrating strong capability for contextual understanding.

## 5.3 ABLATION STUDIES

We conducted ablations to assess the contribution of each component. Soft-label distillation (Table 2) provides clear and consistent gains, particularly for Retrieval and STS, validating the effectiveness of continuous supervision.

Other analyses, including the effect of in-context learning (ICL) and the comparison of negative sampling strategies, showed more nuanced trends: ICL yields marginal or task-specific improvements, while our adaptive margin-based sampling stabilizes training and achieves the best overall performance. For completeness, we report the full tables and detailed results of these analyses in Appendix B.4.

### 5.3.1 ANALYSIS OF SOFT-LABEL DISTILLATION

We investigated the efficacy of our knowledge distillation approach by comparing performance with and without soft labels in Table 2. In this comparison, the configuration"With soft-labeling" uses continuous relevance scores distilled from our teacher pipeline, while the "Without soft-labeling" configuration relies solely on binary hard labels for training.

The results clearly demonstrate the benefits of soft-labeling, with performance improving across nearly all categories and boosting the overall mean score (+0.63). The most significant gains were observed in PairClassification (+2.10) and Retrieval (+1.29), with other notable improvements in tasks such as Summarization and STS. This confirms that the fine-grained, continuous signals from the teacher model provide richer supervisory information than binary labels, enabling the student model to learn more nuanced semantic relationships.

Table 2: Impact of soft-label distillation on model performance across MTEB categories. Using continuous teacher-derived scores (with soft-labeling) improves results compared to only hard labels (without soft-labeling), particularly for retrieval and STS.

| Configuration | Retrieval | Reranking | Clustering | PairClassification | Classification | STS | Summarization | Mean (Task) |
|---|---|---|---|---|---|---|---|---|
| With soft-labeling | 66.18 | 49.13 | 59.25 | 88.67 | 89.97 | 86.69 | 38.93 | 74.12 |
| Without soft-labeling | 64.89 | 48.67 | 59.43 | 86.57 | 89.66 | 86.23 | 38.26 | 73.49 |

## 5.4 ANALYSIS OF CURRICULUM LEARNING

We conduct a series of ablation studies to systematically analyze the impact of curriculum learning (CL) on model performance. Our goal is to understand how the sequence, combination, and blending of different training tasks contribute to building a robust and general-purpose text embedding model.

### 5.4.1 SINGLE-TASK LEARNING RESULTS

We first establish baselines by training on curricula composed of a single task type. This allows us to isolate the effect of each task on the model's capabilities. As shown in the table below, single-task training yields highly specialized models with significant trade-offs.

Table 3: Performance of single-task curriculum. Parentheses indicate the change from the baseline model.

| Strategy | Retrieval | Reranking | Clustering | PairClassification | Classification | STS | Summarization | Mean (Task) |
|---|---|---|---|---|---|---|---|---|
| Baseline | 57.62 | 49.78 | 51.44 | 88.42 | 79.85 | 84.32 | 36.57 | 67.97 |
| STS-only | 61.34 | 48.50 | 59.19 | 86.06 | 89.53 | 86.42 | 38.05 | 72.54 |
| RET-only | 66.84 | 49.07 | 58.85 | 87.74 | 89.85 | 83.57 | 39.21 | 73.43 |

Observation: Single-task training leads to overfitting on narrow task objectives. An STS-only curriculum improves semantic alignment but significantly degrades retrieval capabilities, while a retrieval-only curriculum enhances domain discrimination but fails to capture broader semantic nuances.

### 5.4.2 2-STAGE CURRICULUM RESULTS

Next, we investigate whether a 2-stage curriculum can mitigate the trade-offs observed in single-task training. We find that combining tasks sequentially yields substantial gains, but the order of tasks is critical to the outcome.

Table 4: Comparison of 2-stage curriculum ordering. The $STS \rightarrow RET$ sequence provides the most balanced and significant performance boost.

| Strategy | Retrieval | Reranking | Clustering | PairClassification | Classification | STS | Summarization | Mean (Task) |
|---|---|---|---|---|---|---|---|---|
| Stage1: RET → Stage2: STS | 63.36 | 49.24 | 58.99 | 86.36 | 89.56 | 84.81 | 39.28 | 72.73 |
| Stage1: STS → Stage2: RET | 64.54 | 48.91 | 59.09 | 87.69 | 89.75 | 86.61 | 41.01 | 73.60 |

Observation: All 2-stage curricula outperform single-stage counterparts. Crucially, the task sequence matters: initiating with STS establishes a strong semantic foundation, which is then effectively specialized by retrieval training in Stage 2. The $STS \rightarrow RET$ order proves to be the most robust and synergistic combination.

### 5.4.3 3-STAGE CURRICULUM RESULTS

We further explore expanding to 3-stage curricula by adding a Clustering (CLU) stage, intended to enforce domain-level structure. While the results are mixed, they highlight the challenge of balancing multiple task objectives without catastrophic forgetting.

Table 5: Performance of representative 3-stage curriculum. Task ordering remains the dominant factor.

| Strategy | Retrieval | Reranking | Clustering | PairClassification | Classification | STS | Summarization | Mean (Task) |
|---|---|---|---|---|---|---|---|---|
| CLU → RET → STS | 63.49 | 48.81 | 59.70 | 86.66 | 89.47 | 84.53 | 38.87 | 72.82 |
| RET → STS → CLU | 63.62 | 48.62 | 59.15 | 86.49 | 89.71 | 84.47 | 38.66 | 72.75 |
| STS → RET → CLU | 63.76 | 48.58 | 59.57 | 87.09 | 89.79 | 86.24 | 38.14 | 73.30 |

Observation: Adding a third stage for domain regularization does not consistently outperform the best 2-stage curriculum. The results reinforce that task ordering remains critical, and final-stage fine-tuning (e.g., with STS) can help recover semantic alignment lost in prior stages.

### 5.4.4 IMPACT OF TASK DATA RATIOS IN CURRICULUM AND MULTI-TASK LEARNING

Finally, we investigate how the data mixing ratio between Retrieval (RET) and STS tasks influences performance within both multi-task and curriculum learning frameworks. We experiment with varying STS:RET ratios to identify the optimal data allocation for each strategy.

Table 6: Performance comparison of multi-task learning and curriculum learning under varying STS:RET data ratios. For both strategies, a higher proportion of retrieval data (1:2) yields the best results.

| Strategy | Retrieval | Reranking | Clustering | PairClassification | Classification | STS | Summarization | Mean (Task) |
|---|---|---|---|---|---|---|---|---|
| STS:RET=1:1 | 65.14 | 46.11 | 58.78 | 84.66 | 89.70 | 79.48 | 33.80 | 71.57 |
| STS:RET=2:1 | 64.90 | 48.74 | 59.16 | 86.60 | 89.80 | 84.42 | 36.40 | 73.03 |
| STS:RET=1:2 | 65.71 | 48.90 | 59.19 | 87.29 | 89.77 | 85.57 | 37.79 | 73.57 |
| STS → RET (1:1) | 65.53 | 48.89 | 59.26 | 87.68 | 89.67 | 85.67 | 37.63 | 73.56 |
| STS → RET (2:1) | 65.17 | 49.43 | 59.06 | 87.62 | 89.59 | 85.52 | 36.66 | 73.39 |
| STS → RET (1:2) | **65.63** | **48.50** | **59.24** | **87.58** | **89.76** | **86.45** | **40.01** | **73.81** |

Observation: Our results indicate that the data mixing ratio is a critical hyperparameter. For both multi-task and curriculum learning, a ratio of 1:2 (STS:RET) achieves the highest overall performance, suggesting that the more complex retrieval task benefits from a larger data allocation. Notably, the optimal curriculum setup, STS → RET$(1:2)$, still outperforms the best multi-task equivalent, confirming that a sequential learning path combined with an appropriate data ratio is the most effective strategy.

### 5.4.5 SYNTHESIS AND KEY INSIGHTS

Our granular analysis across single, multi-stage, and mixed-task settings culminates in a clear and actionable insight: how the model learns is as critical as what it learns from. While training on individual tasks yields specialized but brittle models (see Section 5.4.1), a systematically designed sequential curriculum is the key to unlocking synergistic gains across a diverse set of capabilities.

The most significant finding of this analysis is the superior performance of the STS → RET two-stage curriculum. This specific sequence consistently outperforms not only the reverse order but also more complex 3-stage curricula and simultaneous multi-task training (see Sections 5.4.2, 5.4.3, and 5.4.4). We hypothesize that this success stems from an optimal learning progression: the model first builds a robust foundation of general semantic understanding from the STS task, which is then refined and specialized for the high-dimensional, discriminative requirements of retrieval.

Ultimately, this study demonstrates a core principle for model training: a structured, sequential exposure to tasks is more effective than simultaneous, mixed-objective training. The STS → RET curriculum serves as a powerful and principled blueprint for developing versatile embedding models that excel across both semantic similarity and retrieval-centric benchmarks.

## 6 CONCLUSION

In this work, we propose an instruction-based framework for generating high-quality, general-purpose text embeddings from decoder-only language models. The framework combines in-context learning, soft labeling via knowledge distillation from a retrieval teacher, and adaptive hard-negative mining, without requiring either architectural modifications or full fine-tuning.

Empirical evaluation on the MTEB benchmark demonstrates that the proposed approach achieves competitive performance across a wide range of tasks, attaining a top-tier Borda rank. Our curriculum learning analysis further indicates that a structured training strategy—starting with semantic similarity tasks (e.g., STS) followed by retrieval—consistently outperforms both conventional multi-task learning and more complex curriculum variants. These results highlight the importance of task ordering and structure in achieving effective model generalization.

Based on these findings, we identify several directions for future research. The use of a resource-efficient retrieval pipeline for soft-label distillation played a key role in our model's scalability. However, this design also presents a trade-off relative to more computationally expensive supervision from large language models (LLMs). A systematic comparison of these strategies, including both performance and resource efficiency, remains an important open question. In addition, hybrid approaches that integrate the scalability of retrieval-based pipelines with the semantic richness of LLMs may enable the development of embedding systems that better balance efficiency, expressiveness, and generalization across tasks.

## ETHICS STATEMENT

This research does not involve human subjects, personally identifiable information, or sensitive data. All datasets used in our experiments are publicly available and widely adopted in the research community. Our work focuses on methodological advancements in text embedding and retrieval. We acknowledge that, as with other embedding models, potential downstream impacts such as bias amplification or misuse in sensitive applications may arise. We encourage responsible use of our models and adherence to the ICLR Code of Ethics.

## REPRODUCIBILITY STATEMENT

We have made extensive efforts to ensure the reproducibility of our results. All datasets used in our experiments are publicly available benchmarks, including retrieval, classification, clustering, and semantic textual similarity (STS) datasets described in Section 3. Data preprocessing steps and task-specific instruction templates are also fully described in Section 3 and Appendix B. Hyperparameters, training strategies (e.g., curriculum ordering, soft-label distillation, adaptive negative sampling), and model configurations are documented in Section 5 and Appendix B.

Our embedding model and training code are hosted on Hugging Face. However, in order to comply with the double-blind review policy, we cannot disclose the repository link in this submission. The public repository link will be included in the camera-ready version upon acceptance. We also provide random seeds and detailed evaluation processes to facilitate consistent replication. Our experiments were conducted primarily on NVIDIA A100 GPUs with PyTorch 2.1. We believe these resources will facilitate reproducibility and foster future research.

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

## APPENDIX A. THE USE OF LARGE LANGUAGE MODELS

We used a large language model (LLM) solely for text editing and polishing. The LLM assisted in refining grammar, improving readability, and enhancing fluency. It was not involved in research ideation, methodological development, data analysis, or writing of technical content. All scientific contributions are entirely the responsibility of the authors.

## APPENDIX B. ADDITIONAL EXPERIMENTAL DETAILS

### B.1 HYPERPARAMETERS

For parameter-efficient fine-tuning, we employ Low-Rank Adaptation (LoRA) (Hu et al., 2022). The rank is set to 64 and the scaling factor $\alpha$ to 32. We fine-tune the model for a single epoch using a learning rate of $1 \times 10^{-4}$ with a linear decay scheduler and a warm-up ratio of 0.1. We apply gradient clipping with a maximum norm of 1.0 and use a dropout rate of 0.1.

For retrieval-oriented tasks, we use in-batch negative sampling with 7 hard negatives per anchor. Each positive query–document pair is accompanied by multiple sampled negatives, ensuring robust contrastive learning.

Table 7: Detailed hyperparameter settings for Stage 1 and Stage 2 training.

| Parameter | Stage 1 (STS) | Stage 2 (Retrieval) |
|---|---|---|
| Optimizer | AdamW | AdamW |
| Learning rate | 1e-5 | 1e-4 |
| Batch size | 256 | 128 |
| Training epochs | 1 | 2 |
| Warmup ratio | 0.1 | 0.1 |
| Gradient clipping | 1.0 | 1.0 |
| Dropout | 0.01 | 0.01 |
| Learning rate schedule | Linear decay | Linear decay |
| Weight decay | 0.01 | 0.01 |
| Temperature | 0.02 | 0.02 |

### B.2 INSTRUCTION TEMPLATES

Table 8 provides the complete set of task-specific instruction templates used in our training. These instructions are adapted from prior work (Wang et al., 2023; Li et al., 2024) and tailored to align each dataset with its corresponding task objective. For example, retrieval datasets are formatted with query–document pairs, while STS datasets include sentence pairs with graded similarity scores.

### B.3 DETAILS OF DATA CONVERSION

Some datasets were reformulated to fit our unified training scheme. In particular, natural language inference (NLI) pairs were converted into STS-style sentence pairs with graded similarity labels, ensuring compatibility with other semantic supervision tasks. The overall data conversion workflow is illustrated in Figure 3, which shows how premise–hypothesis pairs are mapped into the STS format.

As illustrated in Figure 3, we begin with a collection of NLI sentence pairs labeled as "entailment", "neutral", or "contradiction". Entailment pairs are assigned high similarity scores, contradiction pairs are treated as negative examples with low similarity, and neutral pairs are discarded. The resulting sentence pairs are then reformatted to align with the standard STS input format and incorporated into the training corpus for contrastive learning. Soft similarity scores are further applied, as described in Section 4.1.

Additionally, to handle cases where a single query is associated with multiple positive passages, each query–positive pair is separated into individual training instances. For example, if a query has two positive passages ["A1", "A2"], it is transformed into two examples: (query: "A", positive: "A1") and (query: "A", positive: "A2"). A deduplication step is then applied to remove redundant pairs, which improves training efficiency by reducing unnecessary data duplication.

### B.4 ADDITIONAL ABLATION RESULTS

**Effect of In-Context Learning**  Table 9 reports the detailed comparison of zero-shot, one-shot, and two-shot settings across MTEB tasks. While Summarization benefits from one-shot prompting,

Table 8: Instruction templates for the training datasets used in our experiments

| Task Name | Instructions |
|---|---|
| ArguAna | Given a claim, find documents that refute the claim. |
| ELI5 | Provided a user question, retrieve the highest voted answers on Reddit ELI5 forum. |
| FEVER | Given a claim, retrieve documents that support or refute the claim. |
| FiQA2018 | Given a financial question, retrieve user replies that best answer the question. |
| HotpotQA | Given a multi-hop question, retrieve documents that can help answer the question. |
| MSMARCO | Given a web search query, retrieve relevant passages that answer the query. |
| Natural Question | Given a question, retrieve Wikipedia passages that answer the question. |
| QuoraDupQuestion | Given a question, retrieve questions that are semantically equivalent to the given question. |
| SQuAD | Given a question, retrieve passages that answer the question |
| STS12, STS22, STSBenchmark | Retrieve semantically similar text. |
| AmazonCounterfactualClassification | Classify a given Amazon customer review text as either counterfactual or not-counterfactual. |
| AmazonReviewsClassification | Classify the given Amazon review into its appropriate rating category. |
| Banking77Classification | Given a online banking query, find the corresponding intents. |
| EmotionClassification | Classify the emotion expressed in the given Twitter message into one of the six emotions: anger, fear, joy, love, sadness, and surprise. |
| ImdbClassification | Classify the sentiment expressed in the given movie review text from the IMDB dataset. |
| MTOPIntentClassification | Classify the intent of the given utterance in task-oriented conversation. |
| ToxicConversationsClassification | Classify the given comments as either toxic or not toxic. |
| TweetSentimentExtractionClassification | Classify the sentiment of a given tweet as either positive, negative, or neutral. |
| ArxivClusteringP2P | Identify the main and secondary category of Arxiv papers based on the titles and abstracts. |
| ArxivClusteringS2S | Identify the main and secondary category of Arxiv papers based on the titles. |
| BiorxivClusteringP2P | Identify the main category of Biorxiv papers based on the titles and abstracts. |
| BiorxivClusteringS2S | Identify the main category of Biorxiv papers based on the titles. |
| MedrxivClusteringP2P | Identify the main category of Medrxiv papers based on the titles and abstracts. |
| MedrxivClusteringS2S | Identify the main category of Medrxiv papers based on the titles. |
| RedditClustering | Identify the topic or theme of Reddit posts based on the titles. |
| RedditClusteringS2S | Identify the topic or theme of Reddit posts based on the titles and posts. |
| StackexchangeClustering | Identify the topic or theme of StackExchange posts based on the titles. |
| StackexchangeClusteringP2P | Identify the topic or theme of StackExchange posts based on the given paragraphs. |
| TwentyNewsgroupsClustering | Identify the topic or theme of the given news articles. |
| SciDocsRR | Given a title of a scientific paper, retrieve the titles of other relevant papers. |
| StackOverflowDupQuestions | Retrieve duplicate questions from StackOverflow forum. |

most categories show marginal differences and even slight degradation at two-shot. These results indicate that additional in-context examples may introduce stylistic biases without consistent performance gains.

Table 9: Performance comparison under zero-shot, one-shot, and two-shot in-context learning settings across seven representative MTEB tasks. The results show minimal variation in most tasks, with slight degradation in STS and Summarization under two-shot settings.

| Setting | Retrieval | Reranking | Clustering | PairClassification | Classification | STS | Summarization | Mean (Task) |
|---|---|---|---|---|---|---|---|---|
| 0-shot | 66.46 | 49.18 | 59.20 | 87.82 | 89.61 | 87.24 | 37.86 | 74.14 |
| 1-shot | 66.83 | 49.36 | 59.12 | 88.16 | 89.51 | 85.98 | 41.26 | 74.04 |
| 2-shot | 66.62 | 49.29 | 59.46 | 86.83 | 89.58 | 82.92 | 33.89 | 73.12 |

**Impact of Negative Sampling Strategies**    In Table 10, we compare multiple negative sampling strategies. The adaptive margin-based method (margin = 0.95) consistently outperforms random or fixed top-$k$ sampling. This confirms that filtering out overly similar negatives provides a cleaner contrastive signal and stabilizes training.

Our adaptive margin-based strategy (with a margin of 0.95) achieves the best overall performance. This approach improves upon other methods by dynamically filtering out negatives that are semantically too close to the positive sample, which are often false negatives. By doing so, it provides a clearer contrastive signal and enhances training stability, leading to a more robust final model.

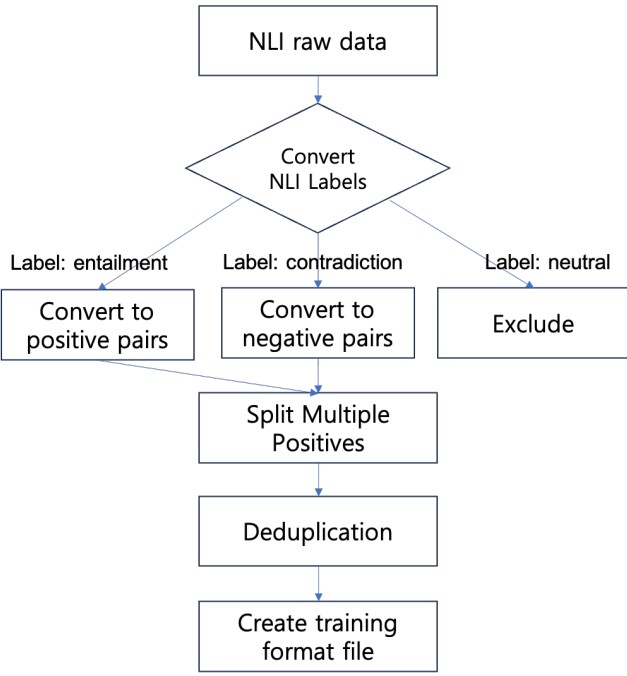

Figure 3: Flowchart illustrating the conversion process from Natural Language Inference (NLI) data to STS-style training examples. Sentence pairs labeled as "entailment" are retained and assigned high similarity scores, while "contradiction" pairs are converted into negative pairs with low similarity scores. Pairs labeled as "neutral" are discarded due to their ambiguous semantic alignment. The resulting examples are reformatted to match the STS input format and used as training data for similarity modeling.

Table 10: Performance comparison of our adaptive margin-based sampling against baseline (in-batch), random sampling, and hard negative (top-k) strategies.

| Strategy | Retrieval | Reranking | Clustering | PairClassification | Classification | STS | Summarization | Mean (Task) |
|---|---|---|---|---|---|---|---|---|
| Baseline (In-batch) | 65.63 | 48.50 | 59.24 | 87.58 | 89.76 | 86.45 | 40.01 | 73.81 |
| Random Sampling | 65.05 | 48.72 | 58.72 | 87.78 | 89.58 | 86.58 | 39.32 | 73.57 |
| Hard Negative (Top-7) | 64.99 | 48.54 | 58.78 | 87.71 | 89.61 | 86.64 | 39.61 | 73.58 |
| Margin-based (margin=0.95) | 67.00 | 49.02 | 58.78 | 87.92 | 89.84 | 86.67 | 41.35 | 74.20 |

