# OpenReview forum: "Unlocking Decoder-LLMs for Text Embedding with Instructions, Soft Supervision and Curriculum Learning"
_ICLR.cc/2026/Conference — Submitted to ICLR 2026_

### Official Review · Reviewer_BD1f · 2025-10-28

**Soundness:** 2
**Presentation:** 2
**Contribution:** 1
**Rating:** 4
**Confidence:** 5

**Summary:**

This paper presents a unified, instruction-based framework that adapts decoder-only Large Language Models into general-purpose text encoders without requiring architectural modifications. The authors propose a four-step learning pipeline: (i) in-context learning to generate context-aware embeddings, (ii) soft supervision through knowledge distillation, (iii) adaptive margin-based hard-negative mining, and (iv) a two-stage curriculum learning strategy (Semantic Textual Similarity (STS) then Retrieval (RET)). The authors then compare their LoRA-finetuned model with this training pipeline against various baselines on the MTEB benchmark, achieving performance on par with the best existing systems.

**Strengths:**

* **State-of-the-Art Performance on MTEB**: The paper introduces an efficient training pipeline that enables decoder-only LLMs to achieve performance on par with the best existing systems on the MTEB benchmark (designed to comprehensively assess the capabilities of sentence representation models across a diverse set of tasks).
* **Thorough Ablation Study**: The authors systematically evaluate the impact of their proposed design choices through comprehensive ablation studies conducted across the full MTEB benchmark. This effectively demonstrates the contribution of each component to the overall performance.

**Weaknesses:**

* **Limited Novelty of Core Techniques**: While the proposed pipeline demonstrates effective integration, its individual components are based on well-established and widely experimented techniques (e.g., in-context learning, knowledge distillation, hard-negative mining). Consequently, the system achieves performance on par with other leading models (that could be explained as many of them also leverage similar foundational training methodologies).
* **Marginal Gains from the Proposed Curriculum Learning**: While the authors emphasize that their two-stage curriculum learning strategy consistently outperforms conventional multi-task learning and more complex curriculum variants, the presented results seem to suggest a relatively small performance improvement. Specifically, the observed performance gap between single-step training (RET-only) and the proposed two-stage curriculum training (STS -> RET) is approximately 0.11 points (as indicated in Tables 3 and 4). This modest gain raises questions about its significance or training noise, especially when compared to the more substantial contributions from other well-established techniques employed in the pipeline, such as soft label distillation (yielding a 0.63 point gain, Table 2) or hard-negative mining (contributing 0.39 points, Table 10).
* **Limited Generalizability to Out-of-Domain Data**: While other leading models also leverage MTEB training data for evaluation, the proposed pipeline training data appears specifically tailored to this distribution while other systems often incorporate a broader range of data sources. This raises concerns about its generalizability and robustness when confronted with out-of-domain data, potentially limiting its practical applicability beyond the MTEB benchmark.
* **Clarity**: The paper seems to suffer from several ambiguities and inconsistencies in its methodological description.
Role of In-Context Learning: The introduction claims the use of in-context learning and few-shot examples to generate specialized embeddings without updating model weights. However, Table 9 contradicts this by showing that the inclusion of few-shot examples consistently decreases the average model performance. Furthermore, the subsequent description of the training pipeline, which involves LoRA finetuning with soft label distillation, obscures the contribution and integration of in-context learning without updating model weights and instruction-following, raising questions about their overall impact and purpose in the final system.
* **Misinterpretation of Hard-Negative Mining Strategy**: In Section 4.3, the authors state they discard "Negative candidates with scores falling below this threshold are excluded during training" (maximum negative score threshold), citing Moreira et al. This directly contradicts the technique described by Moreira et al., which specifically advocates for using candidates below this threshold to effectively avoid the inclusion of false negatives and improve mining efficiency.

**Questions:**

* The pipeline used soft label knowledge distillation, which teacher model(s) is/are used?
* Table 2: soft-labeling give the best model performance of the paper, this model has been also trained with hard negative mining? Two curriculum steps? Only soft-labeling?
* Table 3: What is the baseline model? is it trained or is it the base mistral?
* Concerning Hard-Negative Mining Strategy, is the Author used the strategy from Moreira et al. or is the described one correct (in this case, the citation seems to be irrelevant)?
* This pipeline aims to be computationally efficient. It would be interesting to include the total GPU hours required, and to break down the proportion of GPU usage between the LoRA fine-tuning (to reduce training compute) and the generation of soft labels.
* The primary evaluation was conducted on MTEB, using the model's in-domain training dataset. To further validate the presented results, it would be beneficial to evaluate on out-of-domain distributions using RTEB.

---

### Official Review · Reviewer_nAFZ · 2025-10-30

**Soundness:** 1
**Presentation:** 1
**Contribution:** 1
**Rating:** 0
**Confidence:** 3

**Summary:**

This paper presents a framework for learnign embeddings from LLMs using data selection with BM25 and rerankers and a fine-tuning step. A third step can be added to the pipeline leveraging ICL. The framework is tested using Msitral-7B and evaluated on MTEB(english, v2).

**Strengths:**

As mentionned in the next section, there are more concerns than strengths to this work. See section below.

**Weaknesses:**

**Comments on the format:**

This paper is poorly written and poorly presented, it looks like it abused AI-generation for many reasons, I cite few of them here:
* The related work are really poor, 2 paragraphs only contain 1 citation each, with an abuse of m-dashes. Many papers are also missing concerning the embedding generation with LLMs, like GritLM-7B paper.
* All figure look like they were generated by ChatGPT, no effort was made to improve them. A figure that showcases an example is more appreciated.
* Citations not formated correctly, for example:  *..by Moreira et al. (Moreira et al., 2024),...*, it should be replaced just by *\citet{paperef}*.
* The formatting of Table 1 is random, only the summarization line is bolded.

**Comments on the technical content:**
* The model is only evaluated on English, while other models are multilingual, it is not sure how the proposed approach would perform in a multilingual setting.
* The paper presents a framework for extracting embeddings from LLMs but only shows how it works with Mistral-7B. Other LLMs have been developed since then, Mistral-7B may not be a good reference. Showing that the framework works with other LLMs is needed.
* Authors employ the term "Knowledge Distillation" for describing their sample selection strategy but no KD is really done.
* The method is poorly explained, it's unclear what the "soft label distillation" does during the training. Better detailing and showcasing an example of a step would have helped understanding.

In genreal, the work is very similar to what already has been done with LLMs for representation learning, the novelty and originality of this work is not clear. It needs major updates to match ICLR standards and be published.

**Questions:**

1 - Why not test the framework on other backbone models?

2 - Why limit the evaluation on English only? Did you observe similar improvments on MTEB(multilingual, v2)?

---

### Official Review · Reviewer_2RKu · 2025-10-31

**Soundness:** 3
**Presentation:** 2
**Contribution:** 2
**Rating:** 4
**Confidence:** 3

**Summary:**

This paper proposes to improve general-purpose text embeddings through (i) in-context learning (ICL), (ii) knowledge distillation with soft scores, (iii) curriculum learning, and (iv) hard-negative mining in an adaptive manner. Experiments on the MTEB v2 English demonstrate that the models trained with these methods achieve SOTA results as of June 2025.

**Strengths:**

(1) The paper focus on the important problem of developing general-purpose text embedding models.

(2) The developed model achieves SOTA results on MTEB v2 English as of June 2026.

(3) Detailed ablation studies are conducted on each component of the model.

**Weaknesses:**

(1) The techniques studied in this paper have been explored in prior works, with some methods simply reusing existing approaches without modification. Specifically, the ICL ifollows the same approach as in [1], and hard-negative mining is based on [2]. Knowledge distiallation using soft supervision is also well studied in the literature by applying KL loss using soft reranker scores. A two-stage curriculum learning is also explored in [3], although the detailed config of the curriculum differs.

(2) The two-stage approach (STS followed by Retrieval) is only slightly better than using retrieval data alone (73.60 vs. 73.43). Using 2-shot demonstrations leads to worse performance compared to no demonstrations (73.12 vs. 74.14).

(3) The paper highlights the advantage of avoiding architectural changes and full fine-tuning. However, these characteristics are already common in existing models (e.g., BGE-EN-ICL, Qwen3-Embe, etc.).

(4) There are some missing descriptions. Only the InfoNCE loss is mentioned in the paper. It is not clear how the soft scores are used in the model training.

[1] Making Text Embedders Few-Shot Learners.

[2] Nv-retriever: Improving text embedding models with effective hard-negative
mining.

[3] NV-Embed: Improved Techniques for Training LLMs as Generalist Embedding Models.

**Questions:**

(1) The citation format in the paper is incorrect. There should be a whitespace betwen the text and the citation.

---

### Official Review · Reviewer_9gxN · 2025-10-31

**Soundness:** 2
**Presentation:** 3
**Contribution:** 2
**Rating:** 4
**Confidence:** 4

**Summary:**

The paper proposes a practical recipe to convert a decoder‑only LLM (Mistral‑7B) into a strong, general‑purpose text embedder without architectural changes. The system combines: (i) instruction‑shaped inputs with few‑shot demonstrations and [EOS] pooling to get context‑aware embeddings (§4.2; formatting examples in Table 8, p. 14); (ii) soft supervision via a hybrid retrieval teacher (BM25 + dense retrieval fused with RRF and reranked by a cross‑encoder; Fig. 2 on p. 5) to produce continuous relevance scores (§4.1); (iii) adaptive, margin‑based hard‑negative mining with a “95% of positive” threshold (§4.3, p. 6); and (iv) a two‑stage curriculum that first learns semantic textual similarity (STS), then retrieval (RET) (§5.4, Tables 3–6, pp. 8–9). On MTEB (English v2; 41 tasks) the model reports a mean of 74.12 and states a higher Borda rank than some models with slightly higher mean (Table 1, p. 6). Ablations claim gains from soft labels (Table 2), the STS to RET ordering (Tables 4–6), and the margin‑based mining (Table 10, p. 15). Fine‑tuning uses LoRA; hyperparameters are in §5.1 and Appendix B.1 (Table 7).

**Strengths:**

• Modular training recipe that practitioners can replicate: instruction‑shaped inputs, a hybrid teacher (BM25 + dense + RRF + cross‑encoder), and LoRA fine‑tuning, illustrated in Fig. 2 (p. 5) and Fig. 1 (p. 4).
• Curriculum insight: STS pretraining followed by RET fine‑tuning gives consistent gains over reverse ordering and multi‑task mixing (Tables 4–6, pp. 8–9).
• Ablations: soft‑label on/off (Table 2), ICL shot counts (Table 9), and negative sampling (Table 10). The margin‑based rule is empirically best (Table 10).
• Balanced MTEB performance across categories (Table 1), with standout Summarization (38.93) and strong Retrieval/STS/Pair‑Classification.

**Weaknesses:**

1. Core objective unspecified (major). The paper repeatedly claims “soft labels” but instantiates only InfoNCE, which is typically hard-labeled. To substantiate the thesis, the authors must state and test a genuine soft‑label loss. Also, this choice needs to be properly ablated and justified, as it seems that their core contribution is a general training recipe for text embeddings. Choices of ablations are other objectives that make use of soft labels. For instance: BiXSE (pointwise BCE) [1], LambdaLoss (pairwise BCE with nDCG weighting) [2], RankNet/PairDistill (pairwise BCE) [3-4], and a soft InfoNCE [5] approach.
2. Negative‑mining direction. The text likely flips the inequality; please correct and fully specify top‑K, sampling, and the margin 0.95 used in Table 10.
3. Add more controlled baseline using the same data/mining/curriculum with more strong open base models.
4. Teacher specificity & data hygiene. Name the exact BM25 variant, dense encoder, reranker checkpoint, RRF k (text mentions “typically 60”), and report deduplication against MTEB sources. Greater detail about the data collection and curation process helps with reproducibility and further more with better claims regarding generalization.
5. Novelty concerns. Any individual aspect beyond maybe curriculum has been already examined in great detail.


References
----
[1] BiXSE: Improving Dense Retrieval via Probabilistic Graded Relevance Distillation. Tsirigotis et al. 2025. Treats graded relevance as probabilistic targets and optimizes BCE.
[2] The LambdaLoss Framework for Ranking Metric Optimization. Wang et al. 2018. Metric‑driven pairwise weighting approximating nDCG@k.
[3] Learning to Rank using Gradient Descent (RankNet). Burges et al. 2005. Classic pairwise logistic objective on score differences.
[4] PairDistill: Pairwise Relevance Distillation for Dense Retrieval. Huang et al. 2024. Distills pairwise preferences from a strong reranker into a dense retriever.
[5] Rethinking Negative Pairs in Code Search (EMNLP 2023) proposes Soft‑InfoNCE by re‑weighting negatives; conceptually applicable to graded labels after normalization.

**Questions:**

1. ICL vs LoRA. Which results are “ICL‑only” (no parameter updates) versus LoRA‑tuned? Clarify whether the backbone is frozen and only adapters are trained.
2. Pooling & length. You fix [EOS] pooling; prior work sometimes finds pooling choice important. Add a brief pooling (average, average on top of non-instruction tokens, last token, attention-based pooling) and sequence‑length sensitivity study.

---

### Meta-Review · Area_Chair_UyG7 · 2026-01-07

**Summary:**

The paper suggests a recipe for fine-tuning decoder-only LLMs for text embeddings. The recipe consists of multiple components: a prompt with examples; knowledge distillation via soft supervision; hard negative mining; curriculum training. The authors fine-tune Mistral-7B and obtain good results on MTEB (not the best, but close to the best).

The reviewers mostly criticized the lack of novelty (each individual component in the above recipe has been studied in previous works); lack of clarity in some descriptions (e.g. the distillation procedure was very poorly described); and modest performance gains compared to simpler recipes.

Personally I agree with these criticisms. I also found some parts unclear. And regarding performance gains, the authors' recipe gives them 2nd place (Table 1), but switching off soft labels still keeps them on the 2nd place (Table 2), as does switching off the curriculum (Table 3). So actually switching off all components that the authors emphasize as important, *still leaves them on the same place in the leaderboard*. This makes me confused about why all other models listed in Table 1 are performing worse. I did not find a discussion of this.

**Reviewer Concerns:**

The authors provided responses, but did not upload a revision. The responses clarified some things that were unclear in the submission, but without an updated paper, I don't think any reviewer would have changed their score. And the criticisms about lack of novelty and modest performance gains cannot really be resolved.

**Reviewer Scores:**

Given the above, I think most reviewers would have kept their original scores: 0/4/4/4. While I think score 0 was unfairly low, even discarding it entirely leaves 4/4/4 scores which is unfortunately a consensus in favor of rejection.

---

### Decision · Program_Chairs · 2026-01-26

Reject